East palearctic treefrog past and present habitat suitability using ecological niche models

Andersen Desiree 1
Maslova Irina 2
Purevdorj Zoljargal 3 4
Li Jia-Tang 5
Messenger Kevin R. 6
Ren Jin-Long 5
Jang Yikweon 1 7 jangy@ewha.ac.kr
http://orcid.org/0000-0003-1093-677X Borzée Amaël 8 amaelborzee@gmail.com
1 Department of Life Science and Division of EcoScience, Ewha Womans University , Seoul , Republic of Korea
2 Federal Scientific Center of the East Asia Terrestrial Biodiversity Far Eastern Branch of Russian Academy of Sciences , Vladivostock , Russian Federation
3 Department of Biology, Scholl of Mathematic and Natural Science, Mongolian State University of Education , Ulaanbaatar , Mongolia
4 Department of Forest and Environmental Resources, Chungnam National University , Daejeon , Republic of Korea
5 Chengdu Institute of Biology, Chinese Academy of Sciences , Chengdu , People’s Republic of China
6 Nanjing Forestry University , Nanjing , People’s Republic of China
7 Interdisciplinary Program of EcoCreative, Ewha Womans University , Seoul , Republic of Korea
8 Laboratory of Animal Behaviour and Conservation, College of Biology and the Environment, Nanjing Forestry University , Nanjing , People’s Republic of China
Casazza Gabriele
Electronic publication date: 2022 Mar 3
Publication date: 2022
Volume: 10
Electronic Location ID: e12999
Received 2021 Oct 15; Accepted 2022 Feb 2
Copyright: © 2022 Andersen et al.
Copyright year: 2022
Copyright holder: Andersen et al.
License: This is an open access article distributed under the terms of the Creative Commons Attribution License, which permits unrestricted use, distribution, reproduction and adaptation in any medium and for any purpose provided that it is properly attributed. For attribution, the original author(s), title, publication source (PeerJ) and either DOI or URL of the article must be cited.
License URL: https://creativecommons.org/licenses/by/4.0/

Keywords: Ecological niche model, Past distributions, Dryophytes, Northeast Asia, Treefrog, Yellow sea, East palearctic

Funding: Rural Development Administration of Korea PJ015071022021 Korea Environmental Industry & Technology Institute KEITI 2021002270001 Foreign Youth Talent Program from the Ministry of Science and Technology QN2021014013L Yikweon Jang was funded by the research grants from the Rural Development Administration of Korea (PJ015071022021) and from the Korea Environmental Industry & Technology Institute (KEITI 2021002270001). This work was partially supported by the Foreign Youth Talent Program (QN2021014013L) from the Ministry of Science and Technology to AB. The funders had no role in study design, data collection and analysis, decision to publish, or preparation of the manuscript.

==============================
Ecological niche modeling is a tool used to determine current potential species’ distribution or habitat suitability models which can then be used to project suitable areas in time. Projections of suitability into past climates can identify locations of climate refugia, or areas with high climatic stability likely to contain the highest levels of genetic diversity and stable populations when climatic conditions are less suitable in other parts of the range. Modeling habitat suitability for closely related species in recent past can also reveal potential periods and regions of contact and possible admixture. In the east palearctic, there are five Dryophytes (Hylid treefrog) clades belonging to two groups: Dryophytes japonicus group: Clades A and B; and Dryophytes immaculatus group: Dryophytes immaculatus, Dryophytes flaviventris, and Dryophytes suweonensis. We used maximum entropy modeling to determine the suitable ranges of these five clades during the present and projected to the Last Glacial Maximum (LGM) and Last Interglacial (LIG) periods. We also calculated climatic stability for each clade to identify possible areas of climate refugia. Our models indicated suitable range expansion during the LGM for four clades with the exclusion of D. immaculatus. High climatic stability in our models corresponded to areas with the highest numbers of recorded occurrences in the present. The models produced here can additionally serve as baselines for models of suitability under climate change scenarios and indicate species ecological requirements.

Introduction

Past distributions of species relate to patterns of current distributions, genetics (Carnaval et al., 2009), and biodiversity hotspots (Carnaval & Moritz, 2008; Woodruff, 2010; Tang et al., 2018) and can therefore inform conservation practices. For example, recent genetic analysis of Floreana’s lizards (Microlophus) in the Galapagos supports the hypothesis of “glaciation driven contact” between conspecifics where Floreana and Champion islands were connected during the Pleistocene glacial maximum, and therefore Champion Island may be a source for potential reintroduction of Microlophus to Floreana Island in the event of extinction since they are genetically similar (Torres-Carvajal, Castaño & Moreno, 2021). Similarly, in a study of endemic plants in East Asia, paleogeographic models showed that areas with high climatic suitability in the mid Holocene and last glacial maximum correlated to high levels of endemism in the present, meaning those areas provide long-term stable refugia for relict plant species (Tang et al., 2018).

Climatic stability may indicate areas of climate refugia (Gavin et al., 2014) where a species has been extant consistently across shifting climatic conditions. These areas can harbor the highest levels of species’ genetic diversity and are therefore principally important for conservation (Loera, Ickert-Bond & Sosa, 2017). Identifying areas of high genetic diversity is important for species conservation as genetic diversity promotes adaptation and long-term survival of the metapopulation (Booy et al., 2000; Vandewoestijne, Schtickzelle & Baguette, 2008), particularly in microrefugial populations (Mosblech, Bush & van Woesik, 2011). Climatic stability can also be used to identify areas that would remain suitable for species in different climate change scenarios, which will become vital for persistence of species at risk of extinction (Ashcroft, 2010; Keppel et al., 2012; Morelli et al., 2016). This is especially true for amphibians due to the current extinction crisis in the class (Beebee & Griffiths, 2005; Sodhi et al., 2008; Wake & Vredenburg, 2008). For example, climate change models of a plethodontid salamander (Karsenia koreana) in the Republic of Korea predicted a near total shift in suitable climate for the species under moderate to severe climate change scenarios, wherein the species’ current range would be almost completely outside of the suitable range predicted in the future (Borzée et al., 2019a).

Ecological niche models, also called species distribution or habitat suitability models, represent the presence probability or abundance of species, usually using environmental predictor variables (Guisan & Zimmermann, 2000; Guisan & Thuiller, 2005; Elith & Leathwick, 2009; Guisan, Thuiller & Zimmermann, 2017). These models can be easily projected across geographic ranges and time scales and can therefore be valuable to climate change and paleogeographic research. Such models are particularly useful for species that do not disperse over long distances, such as amphibians which generally have continuous distribution patterns (Duellman, 1999) and do not disperse over long distances (Smith & Green, 2005). In amphibians, paleogeographic suitability models combined with genetic analysis have linked Malagasy bullfrog (Laliostoma labrosum) divergence with periods of high aridity during the Pleistocene (Pabijan et al., 2015).

In the east palearctic (encompassing northeast Asia), treefrogs in the genus Dryophytes face ongoing threats such as habitat loss (Borzée et al., 2015, 2017a; Kuzmin et al., 2017; Borzée, 2020), drought (Kuzmin et al., 2017), pollution (Borzée et al., 2018), invasive species (Wu et al., 2005), disease (Borzée et al., 2017b) and competition (Borzée et al., 2016; Borzée, Kim & Jang, 2016). Among these, habitat loss is considered the biggest risk to the clade, as suitable habitat is converted from artificial wetland (rice paddies) to other forms of agriculture (Kuzmin et al., 2017). Additionally, climate change is expected to negatively impact amphibian species worldwide (Blaustein et al., 2001; Corn, 2005; Lips et al., 2008; Struecker & Milanovich, 2017).

The objectives of the current study were to model present and past habitat suitability and stability for five east palearctic Dryophytes clades: two clades of D. japonicus, D. immaculatus, D. flaviventris and D. suweonensis. To model present and past habitat suitability, we utilized the maximum entropy algorithm (MaxEnt; Phillips et al., 2017) because of its ability to easily create ecological niche models that can be projected to past or future climate scenarios. In creating suitability and stability models for these five clades, we aim to also identify areas of potential climatic refugia and determine climatic preferences for all clades.

Materials and Methods

Focal species

There are currently four recognized species of Dryophytes in the east palearctic: D. japonicus, D. immaculatus, D. flaviventris and D. suweonensis. Dryophytes japonicus is split into two clades (hereafter Clades A and B) with the Clade A being found in Japan northeast of the Chugoku-Kansai border (Dufresnes et al., 2016) and Clade B, likely associated with the name Hyla stepheni (Dufresnes et al., 2016; Borzée et al., 2020b), is found widely across mainland northeast Asia and Japan southwest of the Chugoku-Kansai border (Dufresnes et al., 2016). Dryophytes immaculatus, D. flaviventris, and D. suweonensis are combined into the Dryophytes immaculatus group, with D. immaculatus found in mainland China, D. suweonensis found in the Republic of Korea and the Democratic People’s Republic of Korea, north of the Chilgap mountains, and D. flaviventris found in the Republic of Korea south of the Chilgap mountains and north of the Mangyeong river (Borzée et al., 2020b). The species within the D. immaculatus clade are generally restricted to the low elevation alluvial plains around the Yellow sea (Borzée et al., 2020b) while D. japonicus is found at variable elevation, breeding in a variety of water bodies (Tsuji et al., 2011; Roh, Borzée & Jang, 2014; Borzée et al., 2019b), and human alteration of the landscape resulted in hybridization between the two groups (Borzée et al., 2020a).

Climatic suitability modelling and projection to past climates

We used maximum entropy (MaxEnt ver. 3.4.4; Elith et al., 2011; Phillips et al., 2017) modelling to predict present and past distributions of Dryophytes in the east palearctic, separating the Dryophytes japonicus group into two clades (called A and B in Dufresnes et al., 2016) and the Dryophytes immaculatus group into three species (D. flaviventris, D. immaculatus and D. suweonensis; Borzée et al., 2020b). Occurrences were obtained from the Global Biodiversity Information Facility (GBIF.org, 2021) and non-focal surveys and opportunistic observations conducted by the authors (Supplemental File). In filtering GBIF data, we removed occurrences with more than 4,000 m accuracy, which was the cell size of the environmental layers used to train our models. We omitted occurrences by the National Institute of Ecology Korea (NIEK) since the methodology used by the institute (gridded sampling along transects) differed from other data sources and led to uneven sampling which overestimated common species (D. japonicus) while underestimating rare species (D. flaviventris and D. suweonensis). To further reduce sampling bias, we reduced the occurrences of D. japonicus in the Republic of Korea to 20% using the “Subset” tool in ArcMap 10.8.1 (Esri, Redlands, CA, USA). We finally removed occurrence points of D. japonicus along the Chugoku-Kansai border since the lineage could ambiguously belong to either clade A or B of the species.

When running MaxEnt, we used seven environmental layers (Table 1) with a 2.5 arc-minute (0.0417 decimal degree or ~4 km) resolution. This resolution was used as it was the smallest scale available for the Last Glacial Maximum downscaled climate reconstruction. We used seven independent bioclimatic variables (WorldClim 1.4; Hijmans et al., 2005) with low multicollinearity (Pearson’s r < 0.8) which were previously determined to influence present and past distributions of D. japonicus in the east palearctic (Dufresnes et al., 2016). We opted to only use climatic variables and exclude terrain variables that potentially correlate to present-day distributions (Borzée et al., 2020b) as current topography (e.g., elevation) may not correlate to the same climatic conditions across geologic time scales (Peterson et al., 2011; Jarnevich et al., 2015; Bobrowski et al., 2021). For example, other species have been predicted to shift their elevational distribution as a result of past climate change (Yousefi et al., 2015). For model replication, we used the cross-validate run type option with five replicates (5 k-fold), which uses 80% of occurrences for training and 20% for testing. We selected the option to remove duplicate presence points within the same cell of the environmental layers. To address overfitting (Radosavljevic & Anderson, 2014), we created models at three levels of background points (5,000, 10,000, 15,000) and four levels of randomization multipliers (0.5, 1.0, 1.5, 2.0). We finally projected our trained models to past climates of the last glacial maximum (LGM; 22 kya; WorldClim 1.4; Hijmans et al., 2005) and last interglacial (LIG; 130 kya; WorldClim 1.4; Otto-Bliesner et al., 2006). For our climate projected models, we ran multivariate environmental similarity surface (MESS) analysis (Elith, Kearney & Phillips, 2010) to identify areas of novel past climates with no modern analog where a projected climate suitability model may require more careful interpretation. For each model, we calculated the area under the curve (AUC) for training and test data, true skill statistic (TSS; Allouche, Tsoar & Kadmon, 2006; Lobo, Jiménez-Valverde & Real, 2008), and percent omission of test data for minimum and 10 percent omission thresholds. TSS was calculated from MaxEnt output test and background model values by (from Allouche, Tsoar & Kadmon, 2006):

Table 1 Variables used to train MaxEnt models.

Variable	Description	
Bio1	Annual mean temperature (°C)	
Bio2	Mean diurnal range (°C)	
Bio3	Isothermality	
Bio5	Maximum temperature of warmest month (°C)	
Bio12	Annual precipitation (mm)	
Bio15	Precipitation seasonality	
Bio19	Precipitation of coldest quarter (mm)	
Note:

Bioclimatic variables used to build maximum entropy models for five east palearctic Dryophytes clades.

Sensitivity=aa+c

Specificity=db+d

TSS=Sensitivity+Specificity−1

where:

a = number of occurrences (test points) for which presence was correctly predicted by the model.

b = number of occurrences (background points) for which the species was not found but the model predicted presence.

c = number of occurrences (test points) for which the species was found but the model predicted absence.

d = number of occurrences (background points) for which absence was correctly predicted by the model.

We additionally calculated the difference between training and test AUC for each model. We selected best models for each clade based on these fit statistics, choosing models with low difference between training and test AUC, minimum test omission closest to 0% and 10 percent test omission closest to 10% (or below). To calculate the area of suitability for each clade, we thresholded the models at the maximum test sensitivity plus specificity threshold (true scale statistic or TSS; Allouche, Tsoar & Kadmon, 2006), a threshold which maximizes true presences and absences while minimizing false presences and absences. Finally, to visualize areas of potential climatic refugia, we used the “Cell Statistics” tool (specifying “sum” as the overlay statistics) in ArcMap 10.8.1 (ESRI, Redlands, CA, USA) to create stability models for each clade by calculating the sum of cloglog outcomes of our suitability models across time periods (Loera, Ickert-Bond & Sosa, 2017).

Results

Model evaluation

Models for all clades produced AUC values greater than 0.9 (Table 2), which is considered excellent fit (Dolgener et al., 2014). Models for the D. japonicus Clade B (TSS = 0.8077), D. immaculatus (TSS = 0.7946) and D. flaviventris (TSS = 0.8329) had very good presence-absence prediction capability while models for the D. japonicus Clade A (TSS = 0.9475) and D. suweonensis (TSS = 0.9127) had almost perfect presence-absence prediction capability (Table 2). See Figs. S1–S5 for all tested models and statistics.

Table 2 Model fit statistics.

Clade	Randomization multiplier	Background points	Training AUC	Test AUC	TSS	
Dryophytes japonicus (Clade A)	1.5	5,000	0.9652 ± 0.0034	0.9644	0.9415 ± 0.0050	
Dryophytes japonicus (Clade B)	1.5	10,000	0.9226 ± 0.0101	0.9182	0.7438 ± 0.0453	
Dryophytes immaculatus	1.5	15,000	0.9707 ± 0.0093	0.9632	0.8150 ± 0.0520	
Dryophytes flaviventris	2	15,000	0.9985 ± 0.0004	0.9984	0.9970 ± 0.0004	
Dryophytes suweonensis	2	5,000	0.9910 ± 0.0016	0.9908	0.9820 ± 0.0209	
Note:

Parameters and fit statistics (training and test AUC and TSS) for selected maximum entropy models of five east palearctic Dryophytes clades.

Present and past distributions and areas

The D. japonicus Clade A had a similar suitable range (defined here as the thresholded area of suitability) during the last interglacial (LIG) to its current suitable range (Fig. 1). During the last glacial maximum (LGM), its suitable range shifted south and contracted in the northern part of its current suitable range, with a disjunct suitable range in the Yellow Sea Basin and mainland China. The D. japonicus Clade B also shifted south during the LGM with a LIG suitable range similar to its current suitable range in mainland northeast Asia and southern Japan (Fig. 2). Both D. japonicus clades experienced expansion from the LIG to the LGM, and then subsequent detraction in the present (Table 3). Dryophytes immaculatus experienced suitable range contraction from the LIG to the LGM, but an expansion from the LGM to the present (Fig. 3). Dryophytes flaviventris experienced a suitable range detraction across all three time periods, becoming the clade with the smallest suitable range in the present at 10,760 km2. The clade’s suitable range in the LIG covered large patches in China and the Korean peninsula and had a large area of suitability in the Yellow Sea Basin during the LGM (Fig. 4). Dryophytes suweonensis experienced a marked increase in suitable range from the LIG to the LGM (29,480 km2 to 816,268 km2) mostly in the Yellow Sea Basin (Fig. 5), but in the present its suitable range is similar to its LIG suitable range.

Figure 1 Past and present predicted distribution of Dryophytes japonicus Clade A.

Landscape suitability for Dryophytes japonicus Clade A in the present, last glacial maximum (LGM), and last interglacial (LIG) periods. Past projections are overlaid with multivariate environmental similarity surface (MESS) analysis results representing novel climates with no present-day analog. Habitat stability shows areas with most consistently stable suitable climate for the species over the 130 thousand year period.

Figure 2 Past and present predicted distribution of Dryophytes japonicus Clade B.

Landscape suitability for Dryophytes japonicus Clade B in the present, last glacial maximum (LGM), and last interglacial (LIG) periods. Past projections are overlaid with multivariate environmental similarity surface (MESS) analysis results representing novel climates with no present-day analog. Habitat stability shows areas with most consistently stable suitable climate for the species over the 130 thousand year period.

Figure 3 Past and present predicted distribution of Dryophytes immaculatus.

Landscape suitability for Dryophytes immaculatus in the present, last glacial maximum (LGM), and last interglacial (LIG) periods. Past projections are overlaid with multivariate environmental similarity surface (MESS) analysis results representing novel climates with no present-day analog. Habitat stability shows areas with most consistently stable suitable climate for the species over the 130 thousand year period.

Figure 4 Past and present predicted distribution of Dryophytes flaviventris.

Landscape suitability for Dryophytes flaviventris in the present, last glacial maximum (LGM), and last interglacial (LIG) periods. Past projections are overlaid with multivariate environmental similarity surface (MESS) analysis results representing novel climates with no present-day analog. Habitat stability shows areas with most consistently stable suitable climate for the species over the 130 thousand year period.

Figure 5 Past and present predicted distribution of Dryophytes suweonensis.

Landscape suitability for Dryophytes suweonensis in the present, last glacial maximum (LGM), and last interglacial (LIG) periods. Past projections are overlaid with multivariate environmental similarity surface (MESS) analysis results representing novel climates with no present-day analog. Habitat stability shows areas with most consistently stable suitable climate for the species over the 130 thousand year period.

Table 3 Past and present estimated suitable range area.

Clade	Current range (km2)	Suitability (km2)	
Present	Last glacial maximum	Last interglacial	
Dryophytes japonicus (Clade A)	330,726	537,358	913,731	404,657	
Dryophytes japonicus (Clade B)	958,089	1,475,930	2,411,214	1,614,949	
Dryophytes immaculatus	226,005	1,125,246	873,279	1,470,012	
Dryophytes flaviventris	945	24,840	637,791	1,149,654	
Dryophytes suweonensis	7,853	57,713	270,458	370,161	
Note:

Current estimated range and suitable area (above TSS threshold) based on selectd maximum entropy models for five east palearctic Dryophytes clades during the present (0 kya), last glacial maximum (22 kya) and last interglacial (~130 kya).

For D. japonicus Clade A, the highest climatic stability areas were in lowland and coastal areas of central Japan (Fig. 1), while the highest climatic stability for D. japonicus Clade B occurred in the Korean peninsula and southern Japan (Fig. 2). For D. immaculatus, high climatic stability was found along the Yangtze River (Fig. 3). High climatic stability for D. flaviventris was limited to coastal lowlands in the southwestern corner of South Korea (Fig. 4). Climatic stability for D. suweonensis occurred in the lowlands of the western Korean peninsula, with a small area of stability along the eastern coast of the Democratic People’s Republic of Korea around the city of Wonsan (Fig. 5).

MESS analysis was similar for all clades, with a small area of novel climatic conditions during the LGM in the north and a larger area during the LIG mostly in present-day China (Figs. 1–5, S6). During the LGM, the main novel limiting climatic variables were Bio1, Bio2 and Bio3. During the LIG, the main novel limiting climatic variables for all clades were Bio2, Bio5 and Bio15.

Variable importance

The variables contributing the most to the distribution of the D. japonicus Clade A was Bio19 (precipitation of coldest quarter) at 49.69% (Table 4), while Bio15 (precipitation seasonality) had the highest permutation importance at 84.36% (Table 5). For the D. japonicus Clade B, elevation was the highest for both percent contribution and permutation importance. Elevation was also the highest variable for D. flaviventris, with a contribution of 68.19% and permutation importance of 67.30%. The variable with highest contribution (37.36%) and permutation importance (48.88%) for D. immaculatus was Bio1 (annual mean temperature). For D. suweonensis, elevation had the highest percent contribution at 36.56%, while Bio12 (annual precipitation) had the highest permutation at 76.68%.

Table 4 Percent contribution of environmental variables.

Clade	Bio1	Bio2	Bio3	Bio5	Bio12	Bio15	Bio19	
Dryophytes japonicus (Japanese)	9.29	0.42	0.23	2.50	0.31	14.74	72.51	
Dryophytes japonicus (Mainland)	24.73	2.68	10.73	21.00	6.87	5.10	28.90	
Dryophytes immaculatus	48.27	2.68	2.29	4.31	2.35	3.87	36.22	
Dryophytes flaviventris	31.36	0.12	0.06	18.90	32.50	4.22	12.85	
Dryophytes suweonensis	19.01	0.03	0.01	6.11	41.40	17.02	16.42	
Note:

Variable percent contribution from selected maximum entropy models of Dryophytes clades in the east palearctic. Variables contributing at least 10% to the total model are bolded. See Table 1 for variable explanations.

Table 5 Permutation importance of environmental variables.

Clade	Bio1	Bio2	Bio3	Bio5	Bio12	Bio15	Bio19	
Dryophytes japonicus (Japanese)	8.47	0.75	0.22	0.56	4.02	80.92	5.06	
Dryophytes japonicus (Mainland)	26.41	3.70	9.20	18.47	7.26	4.78	30.20	
Dryophytes immaculatus	67.91	2.81	3.74	8.37	1.20	10.01	5.97	
Dryophytes flaviventris	32.60	0.01	0.00	51.53	14.39	0.53	0.94	
Dryophytes suweonensis	21.89	0.14	0.00	48.74	14.91	7.94	6.37	
Note:

Variable permutation importance from selected maximum entropy models of Dryophytes clades in the east palearctic. Variables contributing at least 10% to the total model are bolded. See Table 1 for variable explanations.

Discussion

Habitat suitability and stability

The highest stability areas for the two D. japonicus clades match their current respective ranges. For the D. japonicus Clade A, this range is in central Japan. The highest stability areas for the D. japonicus Clade B are in the Korean peninsula, between Vladivostok and Khabarovsk along the Ussuri River, and in southern Japan. Comparison with previous research (Dufresnes et al., 2016) suggests a possible correlation between high stability areas and a greater number of haplotypes of Dryophytes japonicus clades in the Korean peninsula and southern Japan. Future research may find a significant correlation between climatic stability and genetic diversity for the two clades. There is also moderately stable area in mainland China west of the Yellow Sea, which makes it surprising that the clade is not found in this region. This may simply be because the species did not disperse to the area while it was accessible, or competition with D. immaculatus or another species (Borzée et al., 2016; Borzée, Kim & Jang, 2016) may have prevented expansion.

High stability areas for D. immaculatus follow the Yellow River Basin where the species is most prevalent. Dryophytes flaviventris has high stability areas in its current range as well as an area south of its range (South Jeolla province of Rep. of Korea). As the species is declining and facing continued habitat destruction, translocation to the South Jeolla province of Rep. of Korea may aid conservation efforts. Additionally, since there is high stability in two coastal areas of Shandong and Hebei, molecular tools or call analysis may be needed to confirm the species in those areas. The highest stability areas for D. suweonensis also match with its current known range with the exception of a high stability area along the east coast of the Democratic People’s Republic of Korea. However, as the species has not been recorded in the area and is geographically isolated from the known range, it is unlikely the species occurs there.

During the LGM, all clades had suitable area in the then-exposed Yellow Sea Basin. This would have allowed for contact and created a situation where admixture was possible. As previous research has estimated the divergence time between D. immaculatus and the D. suweonensis group at around 1.02 mya (Borzée et al., 2020b), it is possible that a minor glaciation during this time (Ehlers, Gibbard & Hughes, 2018) saw the clades splitting from a shared range in the Yellow Sea Basin. However, the merging of all the rivers into the Yellow Sea Basin may have created a barrier that was too difficult for Hylids to cross as observed by the Vistula River segregation of Hyla arborea and Hyla orientalis in Poland (Stöck et al., 2012) and in the Middle East where the Dead Sea Rift has divided Hyla savignyi and Hyla felixarabica (Dufresnes et al., 2019).

Range expansions for the D. japonicus clades, D. flaviventris and D. suweonensis during the LGM are in line with previous studies which have seen LGM expansion in a temperate hylid Hyla sarda (Bisconti et al., 2011). Cooler temperatures that would allow temperate species expansion might have also limited the LGM range of D. immaculatus which is currently found at lower latitudes and more subtropical climates.

Variable responses and species ecology

Variables having the highest percent contribution and/or permutation importance in ecological models indicate potential preference or limiting factors for species. For the D. japonicus Clade A, these variables included precipitation of coldest quarter and precipitation seasonality. Since the region is generally marked by dry winters, this shows a preference towards wetter winters and less precipitation variability between seasons (Fig. 6). This is interesting because a similar preference is not seen in Clade B. Instead, elevation is the variable with the highest contribution and importance, with the species found at low elevations mostly between 0 and 400 m (Fig. 6). Elevation was similarly important for D. flaviventris and D. suweonensis, whose cloglog responses both peak around sea level (0 m). This fits with the species’ habitat preferences for alluvial plains and makes the species particularly vulnerable to sea level rise. For D. suweonensis, the high permutation importance of annual precipitation indicates a preference for moderate annual rainfall with a tolerance for higher rainfall better than lower rainfall and peaking at 1,240 mm. Finally, for D. immaculatus, there is a preference for higher annual mean temperature, which relates largely to its range as the clade present at the lowest latitudes.

Figure 6 Bioclimatic variable response curves for Dryophytes clades.

Bioclimatic variable response curves for the Dryophytes japonicus Clades A and B, D. flaviventris, D. immaculatus and D. suweonensis from maximum entropy models. Responses of each variable are independent of other variables in the models.

Ecological niche modeling also allows for among-species comparisons in responses to environmental variables. Interestingly, annual mean temperature for all clades peaked between 10 to 16 °C indicating preference for warmer climates, with the D. japonicus Clade B tolerating cooler temperatures likely in its northern range in Russia. Maximum temperature of warmest month peaks below 35 °C for all clades likely because of critical thermal limits. A study of another hylid, Dryophytes versicolor, showed a decrease in tadpole speed past about 32 °C with a sharp decrease occurring around 34 °C (Katzenberger et al., 2014).

Conclusions

Because of the numerous threats facing Dryophytes treefrogs in the east palearctic, it is important to determine their ecological requirements in order to implement conservation plans (Park, Park & Borzée, 2021). By using modeling, we have mapped areas of likely refugia, identified areas for potential translocation and determined regional-scale climatic and terrain requirements for five clades of endangered or otherwise at-risk treefrogs in the east palearctic. Additionally, we have provided a baseline for modeling habitat suitability for these clades in future climates under mild to extreme climate change scenarios.

Supplemental Information

Supplemental Information 1 All tested MaxEnt models of Dryophytes japonicus Clade A for present (top left of each panel), last glacial maximum (bottom left of each panel) and last interglacial (bottom right of each panel).

Metrics (top right of each panel) include regularization multiplier, number of background points, training AUC, test AUC, AUC∆, minimum test omission and 10% percent test omission. The selected model is marked by a gold star.

Click here for additional data file.

Supplemental Information 2 All tested MaxEnt models of Dryophytes japonicus Clade B for present (top left of each panel), last glacial maximum (bottom left of each panel) and last interglacial (bottom right of each panel).

Metrics (top right of each panel) include regularization multiplier, number of background points, training AUC, test AUC, AUC∆, minimum test omission and 10% percent test omission. The selected model is marked by a gold star.

Click here for additional data file.

Supplemental Information 3 All tested MaxEnt models of Dryophytes immaculatus for present (top left of each panel), last glacial maximum (bottom left of each panel) and last interglacial (bottom right of each panel).

Metrics (top right of each panel) include regularization multiplier, number of background points, training AUC, test AUC, AUC∆, minimum test omission and 10% percent test omission. The selected model is marked by a gold star.

Click here for additional data file.

Supplemental Information 4 All tested MaxEnt models of Dryophytes flaviventris for present (top left of each panel), last glacial maximum (bottom left of each panel) and last interglacial (bottom right of each panel).

Metrics (top right of each panel) include regularization multiplier, number of background points, training AUC, test AUC, AUC∆, minimum test omission and 10% percent test omission. The selected model is marked by a gold star.

Click here for additional data file.

Supplemental Information 5 All tested MaxEnt models of Dryophytes suweonensis for present (top left of each panel), last glacial maximum (bottom left of each panel) and last interglacial (bottom right of each panel).

Metrics (top right of each panel) include regularization multiplier, number of background points, training AUC, test AUC, AUC∆, minimum test omission and 10% percent test omission. The selected model is marked by a gold star.

Click here for additional data file.

Supplemental Information 6 Novel limiting variables during the last glacial maximum (left) and last interglacial periods (right) for selected models of five eastern palearctic Dryophytes clades.

Click here for additional data file.

Supplemental Information 7 Overfitting statistics for selected models.

Overfitting statistics (AUC ∆, minimum test omission and 10% test omission) for selected maximum entropy models for five east palearctic Dryophytes clades.

Click here for additional data file.

Additional Information and Declarations

Competing Interests

Author Contributions

Data Availability

The authors declare that they have no competing interests.

Desiree Andersen conceived and designed the experiments, analyzed the data, prepared figures and/or tables, authored or reviewed drafts of the paper, and approved the final draft.

Irina Maslova performed the experiments, authored or reviewed drafts of the paper, and approved the final draft.

Zoljargal Purevdorj performed the experiments, authored or reviewed drafts of the paper, and approved the final draft.

Jia-Tang Li performed the experiments, authored or reviewed drafts of the paper, and approved the final draft.

Kevin R. Messenger performed the experiments, authored or reviewed drafts of the paper, and approved the final draft.

Jin-Long Ren performed the experiments, authored or reviewed drafts of the paper, and approved the final draft.

Yikweon Jang conceived and designed the experiments, authored or reviewed drafts of the paper, supervision and funding acquisition, and approved the final draft.

Amaël Borzée conceived and designed the experiments, performed the experiments, authored or reviewed drafts of the paper, and approved the final draft.

The following information was supplied regarding data availability:

Data is available at Mendeley: Andersen, Desiree; Maslova, Irina; Purevdorj, Zoljargal; Li, Jia-Tang; Messenger, Kevin; Ren, Jin-Long; Jang, Yikweon; Borzée, Amaël (2021), “East palearctic treefrog occurrences”, Mendeley Data, V2, DOI 10.17632/k3yysvmv9s.2.

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
