# Peer review of "East palearctic treefrog past and present habitat suitability using ecological niche models"

_PeerJ, doi:10.7717/peerj.12999_

## Round 0.1 · original submission · Major Revisions

Dear Dr. Andersen,

The manuscript presents an interesting research question, and it is well written in professional, unambiguous language. However, the reviewers detect some methodological weaknesses. In particular about the use of only one SDM technique (Maxent) with just default parameters. I agree with them about this issue. A recent article recommends best-practice standards and detailed guidelines to perform analysis (Araújo et al Standards for distribution models in biodiversity assessment. Sci. Adv. 2019; 5 : eaat4858 https://dx.doi.org/10.1126/sciadv.aat4858). I suggest you re-analyse your data at the light of reviewer suggestions and to take into account the article of Araújo to increase adequacy of your models.

So, I encourage you to improve the manuscript according to all tips of reviewers. Please, respond point-to-point to the comments of reviewers to speed up the process of revision.

Once again, thank you for submitting your manuscript to PeerJ and we look forward to receiving your revision.

Sincerely,
Gabriele Casazza

·

Basic reporting

The manuscript is well written in professional, unambiguous language. Field background and literature references are adequate. Figures should be improved in quality for publication. The tables are adequate.

The results section lacks a description of the predicted climate refugia and habitat suitability and stability, which are briefly described in the first two sentences of the first paragraph in their discussion. This information must be detailed in their results section.

Experimental design

This is an original primary research within the Aims and Scope of the PeerJ journal. The research question is well defined, relevant and meaningful. The research seems to have been conducted in conformity with the prevailing ethical standards in the field.

I have some concerns regarding the method used to perform the ENMs, which should be clarified and improved before acceptance. First, although MaxEnt has been widely used for ENMs in the last decades, ensemble models have been highly recommended to reduce uncertainty caused by differences among modeling techniques, improving the accuracy of the fitted models. In addition, MaxEnt treats the occurrence data as "true occurrences", which is likely to be problematic considering their data source (GBIF). Large-scale data aggregators such as GBIF can present much noise in data accuracy, thus requiring special filters and preparation to reduce errors and biased results. According to their methods, they only filtered data from D. japonicus Mainland clade to avoid overabundance and removed duplicate presence points within the same cell of the environmental layers. It is important to check the extent of occurrences found in GBIF, besides evaluating data for spatial outliers. It is important to process the occurrence data to remove erroneous coordinates and spatial outliers/uncertainty.


Also, they are citing Hijmans et al., 2005 as their data source, so I assumed they downloaded the environmental variables from the WorldClim database. However, this information must be stated in their methods section. In addition, it is important to report the version of the data source – in this case, I believe it should be v. 1.4 since v. 2.0 does not present LGM and LIG data available.


Finally, they state that "other parameters were unchanged from MaxEnt's base settings". This sentence requires an explanation of What other parameters they are referring to and why they kept MaxEnt's base settings.

Validity of the findings

All data to reproduce their results are available in public databases (WorldClim - I assumed- and GBIF). Although their methods must be improved, their findings are important to determine potential climate refugia for the analyzed clades, which is essential for conservation actions. Their conclusions are well stated and are linked to the original research question.

Reviewer 2 ·

Basic reporting

The paper is written in a clear and unambiguous way and structured correctly. However, the background/references are insufficient. There is extensive literature using species distribution models in amphibians and frogs in particular to estimate refugia and past areas of distribution that was not cited. Some investigating quite very similar questions in other geographical areas. Although the authors do not need to provide a full literature review or cite everything I suggest I do consider the current background to be insufficient to set up this work in the broad field.
To cite just a few:
Carnaval, A. C. et al. 2009. Stability predicts genetic diversity in the Brazilian Atlantic forest hotspot. - Science 323: 785–789.
Carnaval AC, Moritz C (2008) Historical climate modelling predicts patterns of current biodiversity in the Brazilian Atlantic forest. J Biogeogr 35:1187–1201
Pabijan M, Brown JL, Chan LM, Rakotondravony HA, Raselimanana AP, Yoder AD, Glaw F, Vences M. 2015. Phylogeography of the arid-adapted Malagasy bullfrog, Laliostoma labrosum, influenced by past connectivity and habitat stability. Molecular Phylogenetics and Evolution 92: 11–24.
Paz, A. et al. 2019. Testing effects of Pleistocene climate change on the altitudinal and horizontal distributions of frogs from the Colombian Andes: a species distribution modeling approach. - Front. Biogeogr. 11(1).

There is also extensive literature on the past projections of species distribution models methods and caveats. Again just a few examples here:
Guevara, L., Morrone, J.J. & León-Paniagua, L. (2018a) Spatial variability in species ’ potential distributions during the Last Glacial Maximum under different Global Circulation Models: Relevance in evolutionary biology. Journal of Zoological Systematics & Evolutionary Research, 57, 113–126
Guevara, L. et al. 2019. Spatial variability in species’ potential distributions during the Last Glacial Maximum under different Global Circulation Models: Relevance in evolutionary biology. - J. Zool. Syst. Evol. Res. 57: 113–126.
Qiao, H., Feng, X., Escobar, L.E., Peterson, A.T., Soberon, J., Zhu, G. & Papes, M (2019) An evaluation of transferability of ecological niche models. Ecography, 42(3):521-534. 10.1111/ecog.03986.
Varela, S., Lima-Ribeiro, M.S. & Terribile, L.C. (2015) A short guide to the climatic variables of the last glacial maximum for biogeographers. PLoS ONE, 10, e0129037

Experimental design

No comment

Validity of the findings

Although the field of species distribution models is far from agreeing on the best methods there is widespread agreement on the need to tune the parameters for whatever method is selected. The authors of this study failed to do this by using default Maxent parameters. The use of default parameters is problematic mostly leading to models that are overfit. Overfit models are particularly problematic when they are to be transferred in space or time as is the case in this study.
For tuning of maxent models in particular authors can review the following papers and references within:

RP Anderson, I Gonzalez Jr . 2011. Species-specific tuning increases robustness to sampling bias in models of species distributions: an implementation with Maxent. Ecological Modelling.

JL Brown, JR Bennett, CM . 2017. SDMtoolbox 2.0: the next generation Python-based GIS toolkit for landscape genetic, biogeographic and species distribution model analyses
French – PeerJ.

R Muscarella, PJ Galante, M Soley‐Guardia, RA Boria. 2014. ENM eval: An R package for conducting spatially independent evaluations and estimating optimal model complexity for Maxent ecological niche models. Methods in ecology and evolution.

RP Anderson. 2015. Modeling niches and distributions: it’s not just “click, click, click”. Biogeografía

For transferability issues here are some suggestions:
Qiao, H., Feng, X., Escobar, L.E., Peterson, A.T., Soberon, J., Zhu, G. & Papes, M (2019) An evaluation of transferability of ecological niche models. Ecography, 42(3):521-534. 10.1111/ecog.03986.

Wioletta Werkowska, Ana L. Márquez, Raimundo Real, and Pelayo Acevedo. A practical overview of transferability in species distribution modeling. Environmental Reviews.25(1): 127-133. https://doi.org/10.1139/er-2016-0045

Also the selection of modeling variables is problematic. The authors use topographic variables as predictors in their models. This selection might be good when modeling species distributions on the present (although do remember that species are not responding to elevation but rather variables that vary with elevation). However, when projecting in time the use of topography variables is inappropriate. As species are not responding to elevation itself but other correlates the relationship might not be maintained in time and the projection will thus be biased by the present elevational patterns (e.g. if a species is currently at 2000masl the projected model will try to keep it at the same elevation even if at this elevation temperature will be significantly lower at the LGM). There is again a lot of literature on this topic, here are a few examples:

A.T. Peterson, J. Soberón, R.G. Pearson, R.P.Anderson, E. Martinez-Meyer, M. Nakamura, M.B. Araujo. 2011. Ecological niches and geographic distributions. Princeton University Press

C.S. Jarnevich, T.J. Stohlgren, S. Kumar, J.T.Morisette, T.R. Holcombe. Caveats for correlative species distribution modeling. Ecol. Inform., 29 (2015), pp. 6-15,

Maria Bobrowski, Johannes Weidinger, Niels Schwab, Udo Schickhoff. 2021. Searching for ecology in species distribution models in the Himalayas. Ecological Modelling. 458:109693

Furthermore, since this study is creating temporal projections of species distribution models it is advisable to explore when and where the model is being projected outside the training range of the variables (what conditions in the past are non-analogous to the current climate? In what area is the model then being extrapolated?). For this, one option is to create MESS maps (multivariate environmental similarity surfaces) and use these maps to locate areas where the model should not be trusted , in those areas the model should not be interpreted or at least should be more carefully interpreted.

In lines 146-149 the authors state they used a threshold using TSS which “maximizes true presences and absences”. However, the authors used a presence-background modelling method (Maxent) and only had observations of presence not absences of the species. I wonder why they selected a statistic that is dependent on “True absences”.
For the estimation of refugia the authors say the summed the logistic outputs of their models (also why not the newer clog-log output? ), this implies an area would be more stable with a higher total sum so areas with a higher logistic outputs (e.g 0.8) at the present time and a lower value (say 0.1) at the LGM would have a total of 0.9. But an area with mid prediction (e.g 0.5) in the present and a mid prediction in the past (e.g 0.4) in the past would have exactly the same sum and would evidently be more stable. I wonder if maybe the authors are missing part of the methodology here? If not then this computation seems inappropriate.

Reviewer 3 ·

Basic reporting

This paper deals with species distribution modelling of tree frogs occurring in East Asia. As it stands, I cannot recommend it for publication, for several reasons.

My first concern is that ecological models have already been done on the particular species under focus, including past projections during the last glacial periods (Dufresnes et al. 2016 BMC Ecol Evol for D. japonicus, Borzée et al. 2020 PloS ONE for the D. immaculatus group). Here, the only obvious addition is the analysis of the two main clades of D. japonicus separately. Of course, the authors may have also incorporated some additional localities since 2016 and 2020 (did they?), but I do not see what those updates really brings more to our knowledge of these frogs, especially as their projections are nearly identical to those from the previous papers (which is expected as the same modelling technology was applied = Maxent).

Second, there is a lack of coherence among the delimitation applied here to build models from separate taxa. For the D. immaculatus clade, distinct models are built from D. immaculatus, D. suweonensis and D. flaviventris. However, phylogeographic (minor genetic divergence, widespread admixture) and phenotypic (overlapping morphologies and call properties) studies revealed that all three are one and the same species, the Korean D. suweonensis and D. flaviventris being glacial relics of the Chinese D. immaculatus (Borzée et al. 2020 PLoS ONE). These “species” may thus likely be synonymized in the near future by the community. I understand that this is not a taxonomic paper, but if you want to build separate models for the three glacial lineages (“species”) that compose the D. immaculatus clade, then you should also do separate models for the numerous glacial lineages of D. japonicus. Instead, for D. japonicus, the authors only separated the two main (Miocene-diverged) clades, although both of them feature multiple Plio-Peistocene lineages that all show greater divergence than the Korean “species”. Bottom line, one should stick to a coherent way to build and compare separate models between distinct lineage/species/evolutionary units. In this case, you can either build three models based on three clades that clearly warrants species recognition (D. immaculatus sensu lato and the two main clades of D. japonicus), or about ten models that account for the Pleistocene diversity of each clade (D. immaculatus, D. flaviventris, D. suweonensis, and the ~8 glacial lineages of D. japonicus).

Third, I was a bit confused how the authors use their own and publicly available data to delimit the D. japonicus clades. Both clades occur in Japan, with unclear limits. Did you discard records in areas where we still don’t know the lineage distributions? The paper is rather elusive on the distribution of these clades in general, and even in the way of referring to them, as a “mainland” and “Japanese” clade, even though the mainland clade is also largely found in Japan.

Fourth, and this is more a technical point, did you filter the GBIF dataset in any way? GBIF data often contains misrecords and records with very loose accuracy (dozens of kilometers). I could not find any information on the way you treated the occurrence data, although it is rather important since amphibian occurence largely depends on fine-scale environmental conditions (= microhabitats and microclimates), so inaccurate records are very likely to falsify the reconstructed models.

As it stands, the current manuscript falls short in reporting anything really new in respect to previous studies, plus in my view it misguides readers on the diversity of tree frogs in Asia. That said, the issues I raised can be solved by justifying clearly what novelties are brought here, and by considering a coherent way to build separate models among lineages/species.

Experimental design

See my comments above about the issues to run separate models for some shallow lineages but not others, and the lack of novelty in respect to previous similar studies on these frogs.

Validity of the findings

See my basic reporting above

Reviewer 4 ·

Basic reporting

The introduction is considered to be very brief. It is mainly based on examples of similar articles, rather than on a conceptual presentation of the different topics covered in the paper. There is a large amount of literature to support the use of the main methodological technique used (environmental niche modelling). As well as its use as a valid methodological approach to estimate past distributions.

Experimental design

There are relevant methodological steps that are not clearly explained. There is abundant literature in the ENM world showing that using only one technique (in this case Maxent) can lead to biases in the projections. In this case, this is a critical point, since projections are made in different time windows. For the particular technique used in this work, there are also specific tools that allow for greater validity of the results generated (see Muscarella et al. 2014, Kass et al. 2021).

Kass, J. M., Muscarella, R., Galante, P. J., Bohl, C. L., Pinilla-Buitrago, G. E., Boria, R. A., ... Anderson, R. P. (2021). ENMeval 2.0: Redesigned for customizable and reproducible modeling of species' niches and distributions. Methods in Ecology and Evolution, 12(9), 1602-1608.

Muscarella, R., Galante, P. J., Soley-Guardia, M., Boria, R. A., Kass, J. M., Uriarte, M., & Anderson, R. P. (2014). ENMeval: An R package for conducting spatially independent evaluations and estimating optimal
model complexity for Maxent ecological niche models. Methods in Ecology and Evolution, 5, 1198–1205.

Validity of the findings

no comment

Additional comments

Some minor comments are added that may help to improve the manuscript:

Abstract
56-57 This is an important point that is not addressed in depth in the manuscript, consider removing it.
Introduction
60 A paragraph on the main tool used in the research (ENM) is missing. In addition, the use of ENM as a tool to identify past distributions should be supported.
76-77 There are more up-to-date references for this point
97-98 The expression "we hope" could be changed to a clearer expression of what is expected from the analyses carried out.
Material & Methods
118 By including non-climatic variables in the projections, the concept of climate suitability loses its validity. It is suggested to keep the concept of habitat suitability throughout the text.
124-130 More information on the construction of the occurrence database should be provided. For example, add the date the GBIF data was obtained (including the DOI). Also explain the database found in Mendeley. In this database there are records without source, this should be explained in detail. A table summarizing the different occurrence numbers with which each species is modelled, and the different sources would be very useful.
129-130 Which tool was used for the subsampling, some R library?
143-44 The sources of the climate surface databases should be specified, it is assumed to be Worldclim, which version? 1.2?
145-146 With which tool were the TSS calculated? These are not obtained from the results provided by Maxent.
150 With which methodology and tools was climate stability calculated, was any GIS software used?
Results
156 Please be more cautious with the use of AUC for the evaluation of models. There is a large ENM literature showing that AUC results should be used with caution.

---

## Round 0.2 · accepted · Accept

Dear Dr. Andersen,

The reviewers revised your manuscript and find it now suitable for publication. Despite a reviewer respectfully remaining in disagreement regarding the novelty, s/he recognizes the study is methodologically sound and that the manuscript is well written. So, I am very pleased to inform you that your paper "East Palearctic treefrog past and present habitat suitability using ecological niche models" is accepted for publication in the PeerJ. Congratulations!

Thank you for submitting your work to PeerJ.

Sincerely,
Gabriele Casazza

Reviewer 3 ·

Basic reporting

I cannot really give a different assessment to the new version of the paper. The authors solved some of the issues raised by the other reviewers and myself, but they also prefer to disregard some main concerns raised, about the novelty of the study in respect to previous work, and the taxa/lineages they choose to distinguish in their analyses. Their rebuttal arguments make sense in general, but I do not think they convincingly apply in this particular case.

Regarding replicability, yes, science needs replicability, but replicability is more relevant when different approaches and datasets are used. As recognized by the authors, similar work was carried recently on the same species/populations, using a similar methodology (Maxent).
Regarding the taxa/lineage chosen, the authors supposedly follow currently recognized species (cf. their response) but they nevertheless distinguish two undescribed lineages of D. japonicus. As per my previous review, choosing to delimit taxa simply because they were described even though these arrangements do not follow the same species concept lacks coherence in my view. With a more comparative framework, your study could have tackled some interesting questions regarding niche evolution among the numerous glacial lineages of the D. japonicus/immaculatus diversification.

It seems that we will respectfully remain in disagreement regarding these points. But because the study is methodologically sound, the paper is well written, and as I understand that PeerJ is not about novelty, I am not opposing the acceptance of this paper.

Experimental design

see above

Validity of the findings

see above

Reviewer 4 ·

Basic reporting

The authors have developed the introduction more fully, incorporating points that help to give a better context to their analysis.

Experimental design

In this new version the methods are better explained and detailed.

Validity of the findings

no comment